# Investigating the Interconnection between Environmental, Social, and Governance (ESG), and Corporate Social Responsibility (CSR) Strategies: An Examination of the Influence on Consumer Behavior

**Deinera P. D. Nugroho, Yi Hsu ***, Christian Hartauer and Andreas Hartauer

Department of Business Administration, National Formosa University, Huwei Township, Yunlin 632301, Taiwan; deinerapdn@gmail.com (D.P.D.N.); hartauer.christian@gmail.com (C.H.); andreas.hartauer@europe.com (A.H.)
* Correspondence: yihsu214@gmail.com

**Abstract:** The objective of this research is to develop a structural relational model that examines the relationship between environmental, social, and governance (ESG) variables and corporate social responsibility (CSR). Additionally, the study seeks to determine the impact of these factors on consumer behavior. This research endeavor included the gathering of data from Taiwan and Indonesia, with the aim of investigating the influence of ESG practices and CSR initiatives on customer attitudes and purchase intentions. The study used STATISTA 10 and LISREL to examine the impact of environmental, social, and governance practices on CSR activities. The findings indicated that the integration of environmental practices had a positive effect on CSR initiatives, but the influence of social and governance practices exhibited variability. The research emphasizes the importance of proficiently communicating environmental policies and proposes that organizations should prioritize environmental actions in order to bolster their brand and gain customer confidence. The results also underscore the need for consumer education about the ESG framework. The research closes by emphasizing the management implications, recommending that organizations should embrace a comprehensive strategy towards ESG and CSR policies in order to enhance company performance and ensure long-term sustainability.

**Keywords:** environmental, social, and governance (ESG); corporate social responsibility (CSR); consumer behavior

## 1. Introduction

In the contemporary period, there is a growing consensus on the critical significance of considering several consequential elements while making well-informed business choices. It is becoming apparent that organizations must possess a comprehensive understanding and recognition of the immense importance associated with the integration and management of diverse environmental risks. The observed transition may be attributed to a range of variables, including social and organizational influences, cultural norms and expectations, as well as the aspiration for constructive transformation [1]. The effect on social responsibility may also be attributed to the proactive steps implemented by firms that have acknowledged their significance. Numerous nations have implemented legal frameworks and regulatory measures to institutionalize these endeavors, therefore ensuring that enterprises conduct their operations in a way that aligns with social responsibility principles [2]. The shift in strategy signifies a notable deviation from conventional business methodologies and underscores the increasing acknowledgment of the need for organizations to contemplate their societal influence [3]. For instance, individuals may demonstrate their commitment to philanthropy by actively endorsing and participating in charitable endeavors. They may also advocate for the implementation of equitable social work practices, aiming to ensure fairness and justice.

Additionally, they may prioritize the welfare and well-being of animals. Furthermore, they may cater to the varied dietary preferences and lifestyles of individuals, fostering a sense of unity and inclusivity within communities. Lastly, they may embrace and celebrate the diverse range of cultures and identities that exist among us. The increasing adoption and prioritization of multidimensional issues by firms and investors in their strategies reflect a heightened moral obligation and an expanded sense of ethical awareness [4]. Indeed, there are enterprises that deliberately establish themselves upon and adhere to these fundamental ideas, positioning them as the central tenets of their organizational activities and strategic approaches. The comprehensive and mindful strategy not only caters to and meets the demands of discriminating customers who are becoming more cognizant of these concerns, but it also fosters a deep feeling of satisfaction and accomplishment inside the firm itself, positioning them advantageously in the market [5]. Ultimately, this results in the spontaneous emergence and execution of inventive and progressive approaches that possess qualities of sustainability and social responsibility. Strategies such as Environmental, Social, and Governance (ESG) and Corporate Social Responsibility (CSR) are widely seen in contemporary corporate and commercial environments, playing crucial roles in directing and enhancing the social and environmental initiatives undertaken by firms. These strategies may be used either alone or in combination, depending upon the distinct requirements and goals of the enterprise. Nevertheless, it is crucial to acknowledge that ESG and CSR are not synonymous acronyms. They include separate ideas and theories that play a role in promoting the overall sustainability and ethical behavior of corporations [6]. CSR may be seen as the antecedent of Environmental, Social, and Governance practices, since it has laid the foundation for the adoption of ESG principles in the corporate sphere. CSR is a strategic approach that emphasizes the need of holding corporations responsible for their activities and ensuring that they provide beneficial outcomes for a wide range of stakeholders, such as the environment, customers, workers, communities, and the public domain. The significance of ethical conduct, sustainability, and social responsibility is emphasized. When comparing ESG with CSR, it can be seen that ESG extends the ideas of CSR by offering quantifiable criteria that can be used by investors and customers to assess a company's charitable, social, and internal governance policies. This endeavor transcends the domain of pure altruism and translates these ideas into tangible figures and metrics. This enables stakeholders to have a more comprehensive comprehension of a company's performance with regard to its social and environmental impact.

Indonesia has long recognized the importance of sustainability and has been implementing policies for responsible investment, specifically in the finance industry since 2006. However, it was only in 2014 that Indonesia's Financial Services Authority (OJK) implemented policies to support ESG by creating the Sustainable Finance Roadmap, which consist of two phases of implementation [7]. The first phase, known as Strategic Activities to Implement Sustainable Finance, took place from 2015 to 2019. More recently, in January 2021, Indonesia launched the second phase, which brings together seven components (policy, products, market infrastructure, coordination among related Ministries/Institutions, non-governmental support, human resources, and awareness) into one ecosystem from 2019 until 2024. These policies require listed companies and financial institutions to include ESG disclosure points in their sustainability reports, which are then made available to the public [8]. Additionally, the development of Green Bonds has been introduced to finance or refinance environmentally friendly business activities to protect, restore, and improve environmental quality and function [9].

Similarly, Taiwan's Securities and Futures Bureau (SFB) and the Financial Supervisory Commission (FSC) have recently introduced the Corporate Governance 3.0: Sustainable Development Roadmap and the same roadmap for the certain listed companies in 2020 [10]. These initiatives have been implemented to enhance ESG reporting and Greenhouse Gas (GHG) emission disclosure. As part of these efforts, the Taiwan Stock Exchange (TWSE) and Taipei Exchange (TPEx) have recently reformed their rules to require ESG reports and

the disclosure of ESG performance indicators. Additionally, listed companies are now obligated to issue annual ESG reports based on global reporting guidelines [11].

The phenomenon of heightened awareness first gained traction during the epidemic, but has since transformed into a prevailing norm that is being adopted by a growing number of individuals and organizations [5,12]. There is a growing global focus among citizens and people on the engagement of firms and governments in exhibiting CSR. In light of the rapid occurrence of environmental effects, it is imperative to promptly identify sustainable solutions for a multitude of social, economic, and environmental challenges [2]. According to a survey by Vantage Newsweek, there has been a significant shift in the investment landscape since 2016 with regard to ESG factors. The practice of ESG investing has seen a notable surge in popularity, as shown by the allocation of around 25% of professionally managed assets to ESG strategies by the end of 2016. The anticipated upward trend of this figure may be attributed to the ongoing divestment activities of investors in industries that are deemed contentious, such as tobacco, coal, and armament. As a result, stakeholders now possess elevated expectations about corporations' ability to exhibit enhanced levels of environmental and social responsibility [13]. In contemporary business contexts, it is imperative for organizations to go beyond just reactionary measures and proactively adapt to the ever-changing attitudes, behaviors, and ambitions of customers. In order to optimize their interactions with stakeholders' purchase intention and brand attitude, organizations must actively engage in the identification, isolation, and mitigation of various frictions that may impede these relationships.

This study collected extensive data from two Asian countries, specifically Taiwan and Indonesia, to ensure a comprehensive analysis of the research topic. By gathering data from these diverse locations, the study aimed to capture a wide range of perspectives and insights that could contribute to a more nuanced understanding of the subject matter.

Taking into consideration the aforementioned point, scholarly studies constantly indicate that individuals who engage in consumer behavior (CB) are progressively becoming more aware of the ethical ramifications associated with their buying and financial choices, particularly in regard to sustainability. Corporations are increasingly acknowledging the significance of integrating ESG principles and CSR practices into their organizational activities. Not only does this have implications from an ethical perspective, but it also has the potential to bolster their organizational image, attract clientele, and increase financial outcomes [14]. In the present period of transition towards sustainability and environmental accountability, it is crucial to comprehend the influence of customer behavior on brand attitude and purchase intention. This study endeavors to examine and evaluate the correlation between ESG factors, CSR practices, and customer behavior. The study also aims to acquire a more comprehensive understanding of the intersection of ESG and CSR initiatives, as well as their potential to augment marketing effectiveness via an examination of customer behavior. The objective of this research is to provide a valuable contribution to the current body of information pertaining to ESG and CSR initiatives. The aforementioned analysis offers valuable insights that may be used to enhance corporate decision-making and facilitate the formulation of policies in this particular domain. The objective of this study is to establish a connection between ESG factors and CSR, and to ascertain their influence on customer behavior. Additionally, its objective is to address the deficiency in underexplored methods related to ESG practices and CSR, as well as their influence on customer brand perception and intention to make purchases.

The current study is structured into many chapters, each fulfilling a distinct objective in order to facilitate a thorough and coherent exposition of the findings. Section 1 functions as an introductory section, establishing the framework for the study via the provision of pertinent background information and contextual details essential for comprehending the research subject. In the next chapter (Section 2), an examination will be conducted on the variables included in this study, drawing upon relevant findings from prior research. Section 3 is dedicated to the examination and exploration of the technique used in this study, as well as the subsequent analysis of the collected data. This chapter provides an

overview of the study methodology and methodologies used to examine the association between the variables, along with a description of the data gathering procedure and the analytic techniques performed. In the next chapter (Section 4), the study's results will be presented, shedding light on the influence of ESG factors and CSR initiatives on customer behavior. Section 5 provides a thorough and complete analysis and synthesis of the research findings, leading to a detailed discussion and conclusion. This section presents a concise overview of the main discoveries made in this study, restates the constraints and difficulties encountered throughout the research, and offers suggestions for future investigations in this field.

## 2. Literature Review

### 2.1. Environmental, Social, and Governance (ESG)

The framework system is derived from the notion of responsible investment, which encompasses the strategic and practical integration of ESG aspects into investment decision-making and active ownership [15]. Responsible investing is a guiding philosophy and technique used by investors to evaluate company conduct and anticipate future financial outcomes. This investment idea prioritizes the assessment of firms' sustainable growth and incorporates three essential factors in the analysis and decision-making stages of investments [16]. Through the integration of ESG aspects, investors are given the opportunity to assess the long-term viability and societal implications of corporate operations. ESG embodies an investing concept that seeks to attain consistent long-term value appreciation and functions as a complete, concrete, and practical approach to governance.

Furthermore, an independent organization from the United Nations (UN); The Association for Supporting the SGDs for the UN (ASD) provided an understanding that ESG considerations was once seen as a non-financial factor that are integral to the core management strategies of companies, major financial institutions, and global shareholders. These principles are crucial for sustainable business practices and investments [17].

ESG factors, which encompass the environmental, social and governance aspects of sustainability, do not have a universally agreed-upon definition. Although it is widely understood that ESG factors are the three pillars of sustainability, there is still a lack of agreement in a single definition across international frameworks and standards. This lack of consensus makes it challenging to consistently understand and manage ESG factors. However, the European Banking Authority (EBA) has formulated a concise definition of ESG factors that considers definitions from a number of different institutions. The EBA has formulated a comprehensive definition, that ESG factors are environmental, social, or governance matters that can potentially impact the financial performance or solvency of an entity, sovereign, or individual [18]. The EBA's definition acknowledges that ESG factors can have both positive and negative effects. This means that these factors play a crucial role in assessing opportunities for financial and non-financial entities as they transition towards a more sustainable economy. This aligns with the growing recognition of the need for institutions to adopt a comprehensive, long-term, and strategic approach to address ESG factors.

In this study, the scope of ESG is extensively derived from the EBA, which serves as a valuable point of reference. However, to provide a more specific definition, this study also incorporates definitions from another research that further clarifies the concept. The EBA has meticulously outlined a clear and specific scope for each environmental, social, and governance, providing researchers and readers with a comprehensive understanding of the boundaries associated with ESG. In order to enhance the comprehensibility of ESG, this study will incorporate similar research that stems from the EBA work. This particular research has been carefully selected for its refined nature, aiming to provide a more detailed comprehension of ESG. By relying on these insights, this study aims to enhance the clarity and depth of discussion surrounding ESG, thereby facilitating a more nuanced and informed analysis of this crucial topic.

### 2.1.1. Environmental

The Environmental of ESG are matters that may have potential effects on the financial performance and solvency of entities, sovereign nations, and individuals, i.e., the quality and operation of the natural environment and natural systems are influenced by environmental variables [18]. When it comes to the environment, it encompasses concerns such as the Greenhouse Gas (GHG) emissions, energy consumption and efficiency, air pollutants, water consumption, waste production and management, impact and dependence on biodiversity and ecosystems, as well as innovation in environmentally friendly products and services [19]. The environment indicator refers to how business contributes to the environment. This encompasses various environmental issues as mentioned before, such as GHG, deforestation, and more. There has been a shift in the mindset of business leaders who have recognized the significance of addressing environmental concerns that is not only driven by ethical considerations, but also by the strategic advantage it brings in terms of increasing customer loyalty and driving profits [16]. This can be attributed to the growing environmental awareness among people, who have become more conscious of the significant biological and environmental changes occurring worldwide. As a result, environmental concerns have now become a lucrative source of competitive advantage. Many firms have recognized the potential benefits of embracing eco-friendly practices through their CSR and developing sustainable products to meet the growing demand from conscious consumers [20]. Thus, this finding leads to the following hypotheses:

**Hypothesis 1 (H1).** *Environmental strategies have a positive impact on company's CSR.*

**Hypothesis 2 (H2).** *Environmental strategies have a positive impact on CB.*

### 2.1.2. Social

Social of ESG are the potential influence of social factors on the financial performance and solvency of entities, sovereign nations, and individuals. These social factors may have both positive and negative effects on financial outcomes [18]. Social variables include several aspects that are intricately connected to the fundamental rights, overall well-being, and collective interests of individuals and communities. The characteristics of social factors can be simply illustrated through elements such as workforce freedom of association, child labour, forced and compulsory labour, workplace health and safety, customer health and safety, discrimination, diversity, and equal opportunities, poverty and community impact, effective supply chain management, training and education, customer privacy and the overall impact on the community impacts [19]. A study shows that being responsible towards society greatly affects an organization's performance. It highlights the importance of sharing success with the community. Organizations maintain beneficial relationships by prioritizing employee well-being to boost motivation, productivity, and reduce turnover. Neglecting CSR can harm both the community and the organization, leading to higher rates of unemployment and hindering goal achievement [21].

A finding also revealed that the social dimension had the strongest impact on the brand attitude (BA) and purchase intention (PI) [22]. This suggests that factors related to a company's commitment to diversity, inclusion, and equity have a significant influence on how consumers perceive and engage with a brand. It highlights the importance of considering and addressing social issued in order to effectively enhance brand perception and drive consumer behavior [23]. Hence, based on previous findings, the following hypotheses are developed:

**Hypothesis 3 (H3).** *Social strategies have a positive impact on company's CSR.*

**Hypothesis 4 (H4).** *Social strategies have a positive impact on CB.*

2.1.3. Governance

Governance can be defined as the influence of governance problems on the financial performance or solvency of an organization, nation, or person may be either beneficial or negative [18]. Governance aspects include a range of behaviors related to the governance practices in the business, i.e., practices that pertain to the way a business is managed. Governance can be seen through the establishment of codes of conduct and business principles, accountability, transparency and disclosure practices, fair and transparent executive pay, board diversity and structure, bribery and corruption, stakeholder engagement, and shareholder rights [19]. Taking an example of Diversity, Inclusion, and Equity (DEI) issues in a company, where in the past, companies have demonstrated their support for DEI initiatives through symbolic actions as CSR, yet companies have been hesitant to openly share their DEI metrics. Using ESG metrics is a great excuse to go beyond surface-level statement, e.g., providing diversity scorecard which elevates companies' CSR strategies. Governance metrics in ESG, therefore, presents data on representation and discuss its governance strategies and employee experiences [24]. It has only been briefly proven that Governance dimension of ESG plays the most effective role as compared to the three dimensions on perceived quality [22]. This may be due to the finding that improving governance compliance, such as increasing gender parity or diversifying the board of directors, is easier compared to reduce emissions, which affects ESG score metrics [25]. Thus, referencing this discussion, the following hypotheses are developed:

**Hypothesis 5 (H5).** *Governance strategies have a positive impact on company's CSR.*

**Hypothesis 6 (H6).** *Governance strategies have a positive impact on CB.*

*2.2. Corporate Social Responsibility (CSR)*

CSR refers to the set of principles, policies, and actions adopted by a firm to effectively tackle social, economic, and environmental concerns. Organizations is expected to engage in this commitment voluntarily, without external mandates or regulations imposed by other entities [26]. CSR is cultivated via a synergistic endeavour that encompasses the active participation of leadership, management, and workers. This collective effort serves to establish and maintain the fundamental principles that govern the company's CSR undertakings. However, to effectively implement CSS undertakings, managers are required to carefully select a specific social issue that aligns with their organization's values and goals. This deliberate choice ensures that their efforts and initiatives are focused and impactful, contributing to sustainable social change and positive community impact [21]. There are several effective ways in which companies can contribute positively as part of their CSR initiatives as a response to their chosen social issue [21];

1.  Corporate cause promotions: supporting and raising awareness for social causes;
2.  Corporate social marketing: implementing marketing campaigns to drive behaviour change in society;
3.  Cause-related marketing: making contributions based on product sales;
4.  Corporate philanthropy: making contributions to benefit the community;
5.  Community service: engaging employees in volunteer activities;
6.  Socially responsible business practices: integrating increased social responsibility into day-to-day business operations.

Through the integration of CSR into their operational practices, corporations exhibit their commitment to generating a favourable influence on both society and the environment. The manifestation of this phenomenon is evident in the corporate culture of the organization, which places a high emphasis on upholding ethical principles and making responsible choices. CSR also impacts the organization's stance on volunteering, community investment, and philanthropic initiatives, since it assumes a fundamental role within their comprehensive business plan [27]. Moreover, CSR functions as a guiding framework that facilitates corporations in aligning their activities with the overarching objectives of

sustainable development. It promotes the notion for companies to transcend the exclusive pursuit of financial gains and instead contemplate the societal and ecological ramifications of their activities [28]. By incorporating CSR into their fundamental business operations, organizations may endeavour to achieve sustainable value generation while concurrently making positive contributions to the welfare of society and the environment [29].

Previous studies have shown that customers tend to have a significantly negative reaction towards organisations that demonstrate a lack of social responsibility [21]. This negative perception often leads to a notable decline in the organisation's sales and overall reputation. Therefore, it is imperative for organisations to recognise the potential consequences of ignoring environmental issues and disregarding the limited availability of natural resources. On the other hand, organisations that actively embrace socially responsible practices and demonstrate a genuine commitment to sustainability are much more likely to attract and retain the interest of investors, stakeholders, customers, and general public [14]. This positive perception not only enhances the organisation's brand image but also opens up various opportunities for growth. Henceforth, referencing this discussion, the following hypotheses are developed:

**Hypothesis 7 (H7).** *CSR practices has a positive impact on CB.*

*2.3. ESG vs. CSR*

There is a common misconception that ESG and CSR can be used synonymously and interchangeably. Although these two concepts are related, it is crucial to recognise that these concepts have their own definitive goals and characteristics. ESG has emerged as a framework that build upon the foundation laid by Corporate Social Responsibility. CSR acts as the precursor to ESG, playing a pivotal role in shaping the concept and setting the stage for its evolution [12]. A specific distinction between ESG and CSR lies where CSR generally aims to address a certain social issue based on intrinsic intentions of a company/organisation. The end goal of CSR activities is conducting the activities themselves. On the other hand, ESG is about meeting those intentions and implementing specific sustainable policies. The success or failure of these policies can be measured for the business using ESG ratings [30]. While CSR aims to make businesses accountable for their actions, ESG takes it a step further by introducing measurable criteria to assess and evaluate these efforts that is primarily useful for investors. At its core, CSR is a form of self-regulation that compels companies to actively consider and address their impact on various stakeholders and the broader society. It emphasised the importance of aligning business practices with positive environmental outcomes, consumer well-being, employee welfare, community development, and the overall public sphere. By voluntarily adopting CSR principles, companies demonstrate their commitment to making a positive difference in the world. ESG, on the other hand, expands upon the principles of CSR and transforms them into concrete and quantifiable indicators that are useful for both investors and consumers. It goes beyond the realms of pure philanthropy and presents a comprehensive set of numerical figures that shed light on a company's philanthropic endeavours, social initiatives, and internal governance practices. By providing quantifiable metrics, ESG allows for a more informed understanding of a company's performance in areas of sustainability, ethics, and corporate governance [15]. Overtime, ESG initiatives are deemed compulsory and expected for companies by authorities, yet CSR initiatives has not been enforced by authorities and often left to the discretion of companies [31].

Figure 1 below illustrates the idea between ESG and CSR:

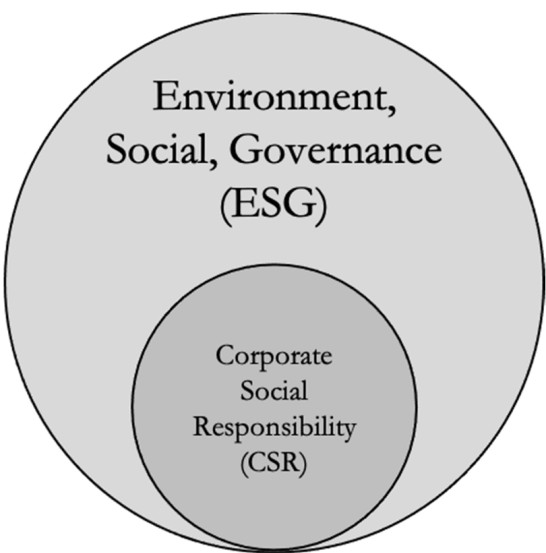

**Figure 1.** Nexus of ESG and CSR.

*2.4. Consumer Behaviour (CB)*

The construct of buy intention has been extensively used in academic research as a reliable indicator of consumer behaviour in relation to later brand purchases. Notably, previous studies have shown a significant correlation between purchase intention and actual consumer behaviour [32]. Brand sentiments have a significant role in forecasting customer behaviour. The formation of brand attitude and its potential impact on significant behavioural outcomes have been thoroughly investigated in previous research [33].

Brand attitude pertains to customers' inclination to express either positive or negative responses towards a brand, which are influenced by their assessments of the product, evaluations of previous purchases, or self-evaluations. The notion was first proposed as a means of comprehending the process through which customers develop attitudes about brands [34]. These views possess the potential to have a substantial influence on consumers' propensity to engage in the purchase of brands [35]. The concept of brand attitude pertains to the inclination of customers to exhibit favourable or unfavourable reactions towards a brand, which are influenced by their assessments of the product, evaluations of previous purchases, or self-assessments [34]. The statement posits that the perception and feeling consumers have towards a brand play a significant role in shaping their purchase choices and brand loyalty [35]. When individuals examine a brand's goods, they develop an attitude via a subjective evaluation of its quality, characteristics, and advantages. Likewise, the construction of attitudes is influenced by consumers' assessments of earlier purchases, encompassing factors such as prior encounters with the brand and contentment with its offerings. Furthermore, the attitudes of customers towards a brand are influenced by their self-evaluations, including personal values, beliefs, and self-image.

The idea of purchase intention is a crucial aspect of consumer behaviour that investigates the inclination of customers to not only make future purchases of a certain brand but also exhibit brand loyalty by refraining from switching to rival brands [36]. The framework paradigm places great importance on the role of perceptual cues in shaping customers' buying behaviour and their desire to make a purchase. In the present model, the purchase choices of customers are influenced by a wide range of circumstances. One issue that plays a significant role is the consumers' impression of the link between the price and quality of the brand. When customers have the perception that a brand provides a positive connection between price and quality, their motivation to make a purchase is increased. Furthermore, buyers may also take into account additional benefits linked to the brand, such as a favourable pricing perception or distinctive attributes that set it apart from rivals. Various elements have a significant role in shaping customers' purchase choices, eventually exerting a profound impact on their enduring brand loyalty. It is important to acknowl-

edge that the cognitive–affective paradigm places significant emphasis on the intricate interaction between perceptual elements and customers' buying behaviour. Through a comprehensive awareness and strategic consideration of these aspects, companies have the ability to successfully influence customers' purchase intentions and foster enduring brand loyalty.

Henceforth, it is important to note that the interaction variables of this study will be visually represented and elucidated in the conceptual model depicted in Figure 2. This model serves as a comprehensive framework that effectively captures and illustrates the various factors and components that are central to the study's investigation.

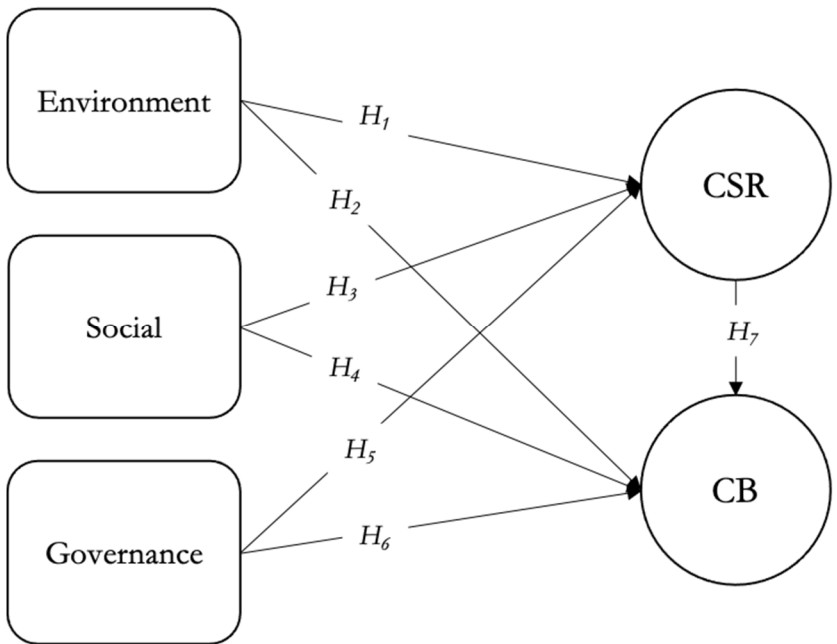

**Figure 2.** Conceptual framework.

## 3. Research Methodology

### 3.1. Variable Operatinalisation and Distribution

Environmental, Social, and Governance (ESG) are the independent variables of this study as separate dimensions. The dependent variables are Corporate Social Responsibility (CSR), and consumer behaviour (CB). The variable operationalisation of this study can be found in Table 1. The objective is to assess the relationship between these factors and determine whether they have a positive or negative impact on each other, based on the respondent's opinions. A positive influence indicates that as one variable increases, the other correspond the same way, while a negative influence suggests that as one variable increases, the other decreases. The interpretation between the individual variables will determine whether the independent variables have a positive/negative impact on the dependent variables.

For instance, based on the findings, environmental strategies according to the respondents showed that it has negative influence on CSR. This indicates that respondents perceive environmental strategies as having detrimental effect on CSR. Meaning that respondent believe that implementing strategies that revolves around environmental concerns does not contribute to a company's CSR efforts, i.e., the perception is that focusing on environmental strategies may not enhance a company's efforts in fulfilling its social responsibility.

**Table 1.** Variable operationalization.

| Factors | Variables | Description | Author |
|---------|-----------|-------------|--------|
| ESG | Environmental | Represents the assessment of a business' contribution to the environment, with a particular emphasis on critical environmental issues. | Dedunu and Sedara (2023) [20] |
| | Social | Encompasses the organisation's relationship with its stakeholders, including employees, suppliers, customers, and the communities in which it operates. | |
| | Governance | Assesses how a firm is governed and managed, which places particular emphasis on company leadership, leadership incentives, stakeholder rights, tax strategies, board structure, and internal controls that aim to foster transparency and accountability. | |
| CSR | CSR | The principles, policies, and actions adopted by a firm to address social, economic, and environmental concerns voluntarily, without external mandates or regulations. | Carroll (1991) [27] |
| CB | Brand attitude | Customers' inclination to respond positively or negatively to a brand, based on their assessment of the product, previous purchases, or self-evaluations. | Burton, Lichtenstein, Netemeyer, Garretson (1998) [34] |
| | Purchase intention | An important aspect of consumer behaviour that studies customers' inclination to continue buying a brand and remain loyal instead of switching to competitors. | Wu, Yeh, and Hsiao (2011) [36] |

### 3.2. Data Collection and Questionnaire

A survey method was used to gain insights into ESG and its impact on CSR, customers' behavior (brand attitudes and purchase intention), and to validate the suggested conceptual model. All of the questionnaire's questions were designed to elicit the most accurate information necessary to accept or reject the proposed model's hypotheses. The respondents are given a set of questionnaires that allows them to agree or disagree in the perception of their opinions and the conceptual result of the interactions between the variables.

The necessity for multidimensional analysis findings necessitates the collection of demographic information.

The rationale for this is because the study offered little hazards to human participants. In accordance with ethical norms, including privacy and confidentiality. The questionnaires include information on the purpose of the survey, instructions on how to respond, and a statement protecting respondents' privacy and confidentiality. The questionnaires have two sections. The first section consists of 40 questions about customers' perspectives on ESG and CSR, as well as their brand attitude and desire to buy. The second half has 8 questions concerning personal information. The characteristics are evaluated using a 5-point Likert scale (fully disagree = 1 and fully agree = 5) ranging from 1 to 5. The questionnaire was originally written in English before being translated into Chinese, and Indonesian. Therefore, respondents from two regions (Taiwan, and Indonesia) can comprehend the questionnaire holistically and react without misunderstanding. The surveys were mostly disseminated using Google Forms to prevent the build-up of inaccurate data, which may otherwise influence the study findings. As a consequence of the qualitative study outcomes and the preliminary survey, the observed variables have been modified and augmented to conform to the research. To obtain the data for hypothesis testing, a non-probabilistic convenience sample was accepted. Consumers over the age of 18 who reside in the three target cultures and have enough reading comprehension are eligible. Initially, 405 participants participated in the survey, but 2 respondents were under the age of 18 and therefore disqualified to fulfil legal requirement of some countries due to ethical codes of conduct. The number of eligible samples was 403, which included 201 respondents from Indonesia, and 202 from Taiwan.

Cronbach's alpha values in each dataset and the pooled data may be used to evaluate the dependability of research constructs.

### 3.3. Data Analysis

#### 3.3.1. Descriptive Statistics

Various techniques are used to showcase the essential attributes of experimentally obtained data, known as descriptive statistics. This serves as an indicator for succinctly displaying quantitative information. This approach proves to be quite beneficial in elucidating and showcasing the many attributes and traits of a certain dataset via a concise and comprehensive summary of the samples. The majority, if not all, levels of mathematics and statistics use focused trend parameters, such as mean, median, and mode, which are the most often utilized versions. There are two primary types of measurement that serve as essential concepts: centralized trend measurement and volatility or dispersion measurement. The former term delineates the fundamental characteristics of a dataset, whereas the later term exemplifies the variability or spread of the data. The demographic statistic is often used to assess both qualitative and demographic aspects. This study used STATISTICA 10 software due to its comprehensive suite of integrated statistical features, including variables such as age, occupation, income, and others.

#### 3.3.2. Linear Structural Relational Model (LISREL)

LISREL, an abbreviation for Linear Structural Relations, is an extensively used software application that has the capability to estimate various covariance structure models. Confirmatory factor analysis is a prominent analytical technique that may be conducted using LISREL. Confirmatory factor analysis (CFA) is a statistical method used by academics to examine and substantiate proposed associations between observable variables and underlying components. Researchers have the ability to examine the adequacy of a proposed model to the data and analyse the magnitude and statistical significance of the associations between variables via the use of LISREL. In addition to the use of confirmatory factor analysis, the LISREL 11 software provides several additional functionalities that facilitate the examination of intricate structural connections within datasets. The use of this approach is prevalent within the realm of research methodology, as it aids researchers in acquiring a deeper understanding of the fundamental framework of the data and facilitates the formulation of conclusions based on empirical evidence.

## 4. Results

### 4.1. Demographic Statistics

The distribution of questionnaires took place between 3 September 2021 and 31 August 2022 via the use of an online survey platform, namely Google Forms. The questionnaires were made available in three languages: Indonesian, Chinese, and English. A total of 403 replies were obtained from the two areas. However, a certain number of participants did not meet the qualifications, while a different number did not comply with the survey requirements. Table 2 presents the demographic characteristics of the data that were obtained. The survey included a total of 201 respondents from Indonesia and 202 respondents from Taiwan. The proportion of individuals living in rural areas, amounting to 72.95% ($n$ = 294), is four times more than the proportion of individuals residing in urban areas, which stands at 27.05% ($n$ = 109). A total of 39.70% of the participants in the study were identified as females, while 60.30% were identified as men. A mere 15.38% of individuals fall into the category of high school students, while a smaller proportion of 6.20% are vocational school students. The remaining majority pursue further education. In terms of marital status, the majority of participants, comprising 78.164% or 315 samples, identified as single. Conversely, married individuals accounted for 21.340% or 86 samples. In relation to age demographics, the cohort of individuals aged between 18 and 25 years accounts for 57.072% (230 observations) of the whole sample. The data indicates that individuals with the lowest income, namely less than USD 250 per month, exhibit the greatest rate of 42.9%

(173 samples). Conversely, those with the highest income, exceeding USD 5,000 per month, constitute the smallest percentage, accounting for just 0.7% (three samples). The complete data obtained from the conducted surveys are shown in Table 2. The assessment of the test items' reliability and internal consistency was conducted using the Cronbach's alpha test. A coefficient $\alpha$ of 0.700 is often acknowledged as indicative of poor dependability, but a coefficient $\alpha$ of 0.800 is seen to be sufficient. In this work, a generally used criterion of 0.800 is utilized to ascertain the dependability of the calculated variables [37]. The findings shown in Table 3 indicate that the Cronbach's alpha test produced values beyond the established threshold, hence suggesting a considerable degree of dependability. As a result, the expectation of the rest designer to extract interpretable statements from the collection of elements was justified [38].

**Table 2.** Descriptive statistics of samples (N = 403).

| Description | Total | | Taiwan | | Indonesia | |
|---|---|---|---|---|---|---|
| Nationality | 403 | (100%) | 201 | (49.876%) | 202 | (50.124%) |
| Area | 403 | (100%) | 201 | (100%) | 202 | (100%) |
| Countryside | 109 | (27.047%) | 59 | (29.353%) | 50 | (24.752%) |
| City | 294 | (72.953%) | 142 | (70.647%) | 152 | (75.248%) |
| Gender | 403 | (100%) | 201 | (100%) | 202 | (100%) |
| Female | 243 | (60.298%) | 130 | (64.677%) | 113 | (55.941%) |
| Male | 160 | (39.702%) | 71 | (35.323%) | 89 | (44.059%) |
| Education | 403 | (100%) | 201 | (100%) | 202 | (100%) |
| Highschool Grad | 62 | (15.385%) | 16 | (7.970%) | 46 | (22.772%) |
| VoTech Program [1] | 25 | (6.203%) | 2 | (0.995%) | 23 | (11.386%) |
| Bachelor's Degree | 267 | (66.253%) | 144 | (71.642%) | 123 | (60.891%) |
| Master's Degree | 47 | (11.663%) | 37 | (18.408%) | 10 | (4.950%) |
| Doctoral Degree | 2 | (0.496%) | 2 | (0.995%) | 0 | (0%) |
| Marital status | 403 | (100%) | 201 | (100%) | 202 | (100%) |
| Single | 315 | (78.168%) | 189 | (94.030%) | 126 | (62.376%) |
| Married | 86 | (21.340%) | 12 | (5.970%) | 74 | (36.634%) |
| Divorced | 0 | (0%) | 0 | (0%) | 0 | (0%) |
| Widow | 2 | (0.496%) | 0 | (0%) | 2 | (0.990%) |
| Age | 403 | (100%) | 201 | (100%) | 202 | (100%) |
| Under 18 | 2 | (0.496%) | 0 | (0%) | 2 | (0.990%) |
| 18–25 | 230 | (57.072%) | 134 | (66.667%) | 96 | (47.525%) |
| 26–35 | 90 | (22.333%) | 50 | (24.876%) | 40 | (19.802%) |
| 36–45 | 24 | (5.955%) | 11 | (5.473%) | 13 | (6.436%) |
| 46–55 | 34 | (8.437%) | 6 | (2.985%) | 28 | (13.861%) |
| Over 55 | 23 | (5.707%) | 0 | (0%) | 23 | (11.386%) |
| Occupation | 403 | (100%) | 201 | (100%) | 202 | (100%) |
| Student | 221 | (54.839%) | 139 | (69.154%) | 82 | (40.594%) |
| Company employee | 99 | (24.566%) | 45 | (22.388%) | 54 | (26.733%) |
| Civil servant | 16 | (3.970%) | 0 | (0%) | 16 | (7.921%) |
| Self-employed | 53 | (13.151%) | 17 | (8.456%) | 36 | (17.822%) |
| Homemaker | 13 | (3.226%) | 0 | (0%) | 13 | (6.436%) |

**Table 2.** *Cont.*

| Description | Total | | Taiwan | | Indonesia | |
|---|---|---|---|---|---|---|
| Retired | 1 | (0.248%) | 0 | (0%) | 1 | (0.495%) |
| Chairman | 0 | (0%) | 0 | (0%) | 0 | (0%) |
| Monthly income | 403 | (100%) | 201 | (100%) | 202 | (100%) |
| Less than USD 250 | 173 | (42.928%) | 78 | (38.806%) | 95 | (47.030%) |
| USD 251–USD 500 | 109 | (27.047%) | 48 | (23.881%) | 61 | (30.198%) |
| USD 501–USD 1000 | 50 | (12.407%) | 19 | (9.453%) | 31 | (15.347%) |
| USD 1001–USD 2500 | 52 | (12.903%) | 46 | (22.886%) | 6 | (2.970%) |
| USD 2501–USD 5000 | 16 | (3.970%) | 7 | (3.483%) | 9 | (4.455%) |
| More than USD 50,000 | 3 | (0.744%) | 3 | (1.493%) | 0 | (0%) |

[1] Vocational–technical school.

**Table 3.** Construct reliability (Cronbach's Alpha).

| Constructs | Pooled | Taiwan | Indonesia |
|---|---|---|---|
| Environmental | 0.936 | 0.936 | 0.882 |
| Social | 0.888 | 0.888 | 0.906 |
| Governance | 0.988 | 0.980 | 0.998 |
| CSR | 0.905 | 0.905 | 0.977 |
| CB | 0.923 | 0.923 | 0.988 |

Notes: CSR = corporate social responsibility; CB = consumer behavior.

*4.2. LISREL Testing*

4.2.1. Covariance Matrix

The purpose of doing a covariance matrix analysis is to assess the degree of association between each observed variable [39]. According to the findings presented in Table 4, the datasets for pooled, Taiwan, and Indonesia all had positive values. This suggests a statistically significant positive link between the variables being examined.

**Table 4.** Covariance matrix of the measured variables.

| Pooled | CSR | CB | E | S | G |
|---|---|---|---|---|---|
| CSR | 1.001 | | | | |
| CB | 0.714 | 0.870 | | | |
| E | 0.704 | 0.634 | 0.826 | | |
| S | 0.674 | 0.586 | 0.660 | 0.781 | |
| G | 0.811 | 0.709 | 0.762 | 0.669 | 0.971 |
| **Pooled** | **CSR** | **CB** | **E** | **S** | **G** |
| CSR | 1.115 | | | | |
| CB | 0.829 | 0.941 | | | |
| E | 0.832 | 0.770 | 0.914 | | |
| S | 0.735 | 0.638 | 0.686 | 0.765 | |
| G | 0.984 | 0.873 | 0.880 | 0.723 | 1.166 |
| **Pooled** | **CSR** | **CB** | **E** | **S** | **G** |
| CSR | 1.115 | | | | |
| CB | 0.587 | 0.797 | | | |
| E | 0.832 | 0.480 | 0.914 | | |
| S | 0.735 | 0.516 | 0.686 | 0.765 | |
| G | 0.984 | 0.528 | 0.880 | 0.723 | 1.166 |

Notes: CSR = corporate social responsibility; CB = consumer behaviour; E = environmental; S = social; G = governance.

### 4.2.2. Goodness-of-Fit Testing

The chi-square ($\chi^2$) goodness-of-fit test was used in this research, a commonly utilized statistical method for this particular sort of investigation. The primary aim of this study is to ascertain if the observed distribution can be effectively approximated by a certain probability distribution. Furthermore, the chi-square test is used to assess the homogeneity of variances computed for numerous samples obtained from a population that conforms to a normal distribution [40]. As the degrees of freedom (df) approach infinity, the chi-square distribution converges to a normal distribution. The findings shown in Table 5 indicate that the chi-square to degrees of freedom ratio ($\chi^2$/df) is 0, indicating a state of total independence between the variables and the absence of any association. The findings obtained from the chi-square test were found to be statistically significant, as shown by a *p*-value (P) that was less than 0.05 [41]. The chi-square analysis also indicated that the root-mean-square error of approximation (RMSEA) value is 0.00, which suggests an excellent fit since values below 0.01 are considered indicative of a good fit. It can be determined that the measurement model in this research satisfies all the necessary requirements, including a *p*-value of 1.00 for the model chi-square, an RMSEA of 0.00, and 0 degrees of freedom [42]. Consequently, it can be concluded that the model is saturated and exhibits a perfect fit.

**Table 5.** Goodness-of-fit statistics measurement.

| Fit Measures | Pooled | Taiwan | Indonesia |
|:---:|:---:|:---:|:---:|
| $\chi^2$ | 0.00 | 0.00 | 0.00 |
| P | 1.00 | 1.00 | 1.00 |
| RMSEA | 0.00 | 0.00 | 0.00 |

Note: $\chi^2$ = chi-square; P = *p*-value; RMSEA = root-mean-square error of approximation.

### 4.2.3. Hypotheses Testing

The results of hypothesis testing for each dataset are shown in Tables 6–8. Figures 3–5 will depict the model representing the outcomes of the hypothesis testing. In order to enable a comparative analysis and provide a comprehensive view of the findings, the datasets are consolidated into a single table, as seen in Table 9. The results demonstrate disparities in the null hypotheses that were rejected across the two nations and the aggregated dataset, warranting additional examination and analysis.

The combined findings derived from the data of both nations demonstrate that the perceived ESG factors have a favourable and statistically significant impact on CSR and CB. This outcome lends credence to Hypotheses 2–6, respectively. Nevertheless, the influence of environmental indicators on CSR is shown to be statistically negligible, hence failing to support hypothesis H1. Furthermore, it can be shown that CSR has a beneficial influence on CB, therefore providing support for hypothesis H7. The summary of the hypotheses testing findings is shown in Table 6, while the model illustrating the hypotheses testing results may be seen in Figure 3.

**Table 6.** Hypotheses testing—LISREL result from pooled data.

| Independent Variables | Dependent Variables | Hypotheses | Est | t-Value | Supported (Yes/No) |
|:---:|:---:|:---:|:---:|:---:|:---:|
| E | CSR | H1 | 0.111 | 1.732 | No |
| S | CSR | H2 | 0.313 | 5.802 | Yes |
| G | CSR | H3 | 0.532 | 10.136 | Yes |
| E | CB | H4 | 0.181 | 2.671 | Yes |
| S | CB | H5 | 0.122 | 2.135 | Yes |
| G | CB | H6 | 0.258 | 4.298 | Yes |
| CSR | CB | H7 | 0.295 | 5.798 | Yes |

Notes: Supported: Yes (Est. > 0 and ｜t-value｜ ≥ 1.96). CSR = corporate social responsibility; CB = consumer behaviour; E = environmental; S = social; G = governance.

**Table 7.** Hypotheses testing—LISREL data result from Taiwan.

| Independent Variables | Dependent Variables | Hypotheses | Est | t-Value | Supported (Yes/No) |
|---|---|---|---|---|---|
| E | CSR | H1 | 0.181 | 2.303 | Yes |
| S | CSR | H2 | 0.314 | 4.488 | Yes |
| G | CSR | H3 | 0.512 | 8.271 | Yes |
| E | CB | H4 | 0.331 | 4.181 | Yes |
| S | CB | H5 | 0.087 | 1.192 | No |
| G | CB | H6 | 0.291 | 4.086 | Yes |
| CSR | CB | H7 | 0.182 | 2.579 | Yes |

Notes: Supported: Yes (Est. $> 0$ and $|\text{t-value}| \geq 1.96$). CSR = corporate social responsibility; CB = consumer behaviour; E = environmental; S = social; G = governance.

**Table 8.** Hypotheses testing—LISREL data result from Indonesia.

| Independent Variables | Dependent Variables | Hypotheses | Est | t-Value | Supported (Yes/No) |
|---|---|---|---|---|---|
| E | CSR | H1 | 0.181 | 3.282 | Yes |
| S | CSR | H2 | 0.314 | 6.395 | Yes |
| G | CSR | H3 | 0.512 | 11.786 | Yes |
| E | CB | H4 | −0.064 | −0.844 | Yes |
| S | CB | H5 | 0.508 | 7.235 | No |
| G | CB | H6 | −0.062 | −0.900 | No |
| CSR | CB | H7 | 0.294 | 4.318 | Yes |

Notes: Supported: Yes (Est. $> 0$ and $|\text{t-value}| \geq 1.96$). CSR = corporate social responsibility; CB = consumer behaviour; E = environmental; S = social; G = governance.

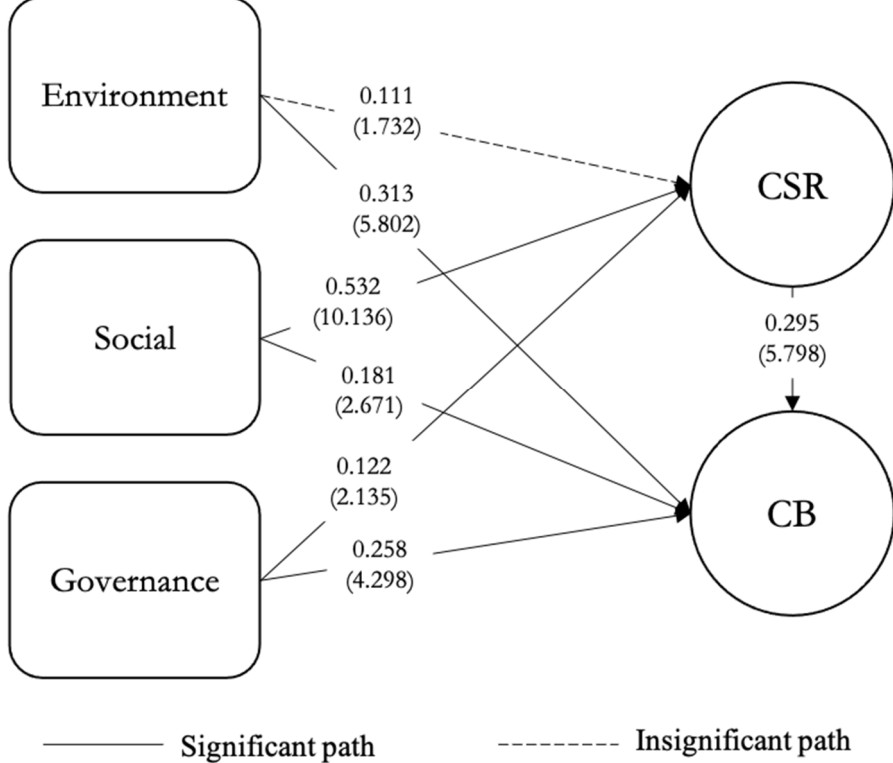

**Figure 3.** LISREL result model of pooled data.

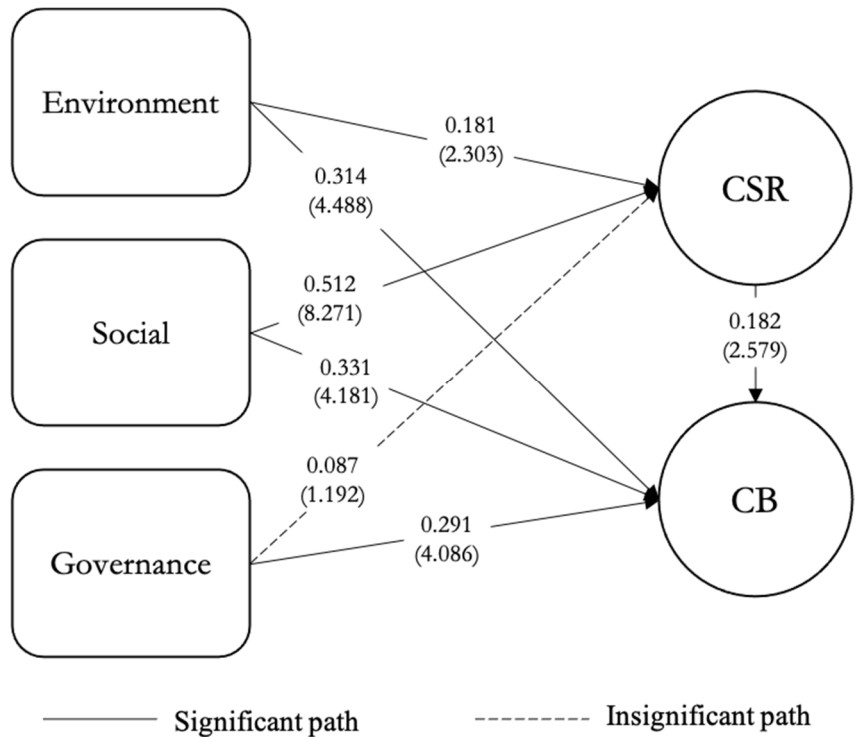

**Figure 4.** LISREL data result model from Taiwan.

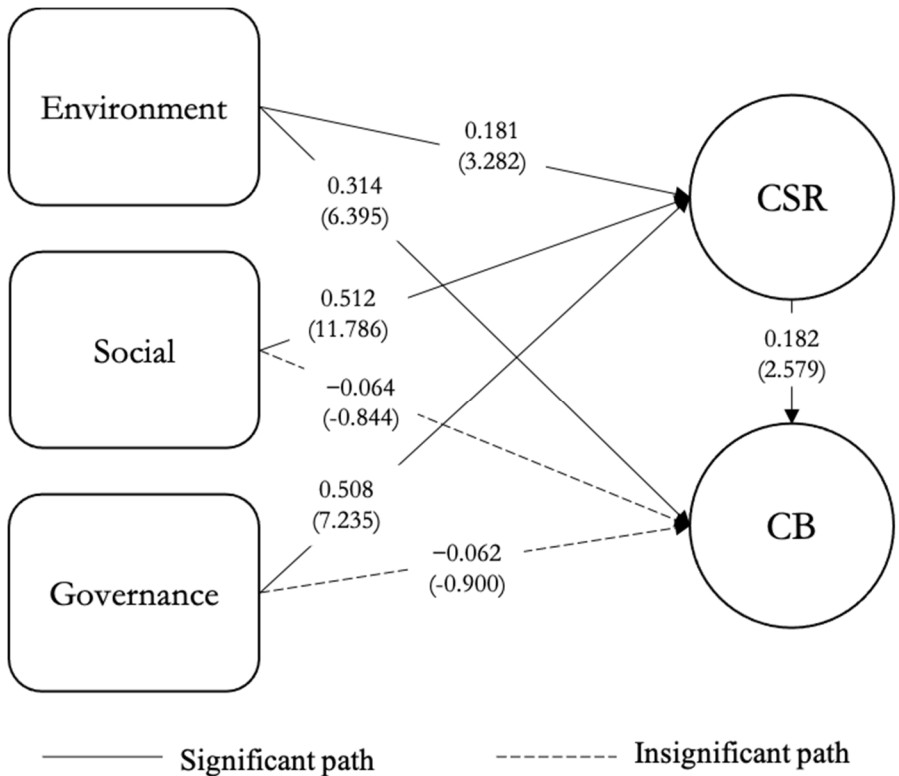

**Figure 5.** LISREL data result model from Indonesia.

**Table 9.** Hypotheses testing—LISREL result from the three sets of data.

| Hypotheses | Supported (Est.) | | |
|---|---|---|---|
| | Pooled | Taiwan | Indonesia |
| H1: E → CSR | No | Yes | Yes |
| | 0.111 * | 0.181 | 0.181 |
| H2: S → CSR | Yes | Yes | Yes |
| | 0.313 | 0.314 | 0.314 |
| H3: G → CSR | Yes | Yes | Yes |
| | 0.532 | 0.512 | 0.512 |
| H4: E → CB | Yes | Yes | No |
| | 0.181 | 0.331 | −0.064 * |
| H5: S → CB | Yes | No | Yes |
| | 0.122 | 0.087 * | 0.508 |
| H6: G → CB | Yes | Yes | No |
| | 0.258 | 0.291 | −0.062 * |
| H7: CSR → CB | Yes | Yes | Yes |
| | 0.295 | 0.182 | 0.294 |

Notes: * indicates | t-value | $\leq$ 1.96. Supported: No (Est. < 0 and | t-value | $\leq$ 1.96). E = environmental; S = social; G = governance; CSR = corporate social responsibility; CB = consumer behavior.

The results obtained from the data collected in Taiwan indicate that the perceived ESG factors have a positive and statistically significant influence on CSR and CB. These findings provide confirmation for Hypotheses 1–4 and 6, respectively. However, the influence of social indicators on consumer behaviour is shown to be statistically negligible, leading to the rejection of H5. Furthermore, the empirical evidence suggests that CSR has a significant beneficial impact on CB, therefore providing support for H7. Table 7 provides a succinct overview of the hypotheses testing outcomes, while Figure 4 visually presents the model representing the hypotheses testing findings obtained from the Taiwanese context.

The results obtained from the study conducted in Indonesia provide evidence that the perceived ESG variables have a favourable and statistically significant impact on CSR, hence verifying the hypotheses H1 and H2. Furthermore, it is worth noting that H3 is also a significant factor to consider. In contrast to Taiwan, the social variables had a noteworthy and constructive influence on CB, hence providing support for hypothesis H5. In contrast, the impact of environmental and governance indicators on CB is shown to be both negative and statistically negligible. Consequently, the hypotheses H4 and H6 are not substantiated. Moreover, it is hypothesized that CSR has a good and substantial impact on the supporters of CB, as stated in H7. Table 8 provides a succinct overview of the outcomes of the hypotheses testing, while Figure 5 visually presents the model depicting the hypothesized test findings obtained from the Indonesian database.

## 5. Discussion and Conclusions

### 5.1. Summary of the Research Result

The next section will be separated into two segments, analysing the aggregated data from Taiwan and Indonesia as a comprehensive depiction of a certain region in Asia, and examining the data independently for each nation. The project intends to address the current vacuum in knowledge on untapped ESG and CSR measures, and their specific influence on customer behaviour. Furthermore, this research aims to acquire a more comprehensive and nuanced comprehension of the intersection of ESG and CSR strategies, as well as their potential synergistic impact on marketing success, by a thorough examination of customer behaviour. The study results indicate that the influence of ESG initiatives on a company's CSR is not uniform across all initiatives. This means that, according to the result obtained from the perceptions of the respondents, from the three pillars of ESG, only a part of the pillars has significant influence on the perception of a company's CSR. The impact of environmental practices within the ESG framework on a company's CSR is shown to be low [22]. One plausible explanation for this result is that firms may not publicly disclose

their environmental activities. corporations may choose to do so as a result of worries associated with "greenwashing", a phenomenon that has prompted several corporations to refrain from publicly disclosing their environmental endeavours [43].

Additionally, the influence of social and governance indicators on a company's CSR is considerable. Moreover, it is worth noting that both ESG policies and CSR initiatives have a significant impact on customer behaviour in a favourable manner. This implies that customers have a greater propensity to react favourably and make purchases from firms that adopt ESG and CSR policies.

The study conducted in Taiwan indicates that there is no substantial impact of the governance factor on CSR. The observed outcome may be ascribed to the preponderance of family-owned enterprises in Taiwan, whereby decision-making authority is concentrated within the familial sphere, leading to restricted power for shareholders and delayed availability of corporate information. The aforementioned scenario underscores a significant deficiency in the corporate governance of Taiwanese companies, suggesting that the significance of governance procedures in the context of corporate social responsibility is not well recognized [44]. However, it is important to note that the research also emphasizes the substantial influence of a company's engagement with stakeholders and its dedication to environmental attributes on its CSR. This, in turn, has a favourable effect on consumers' brand attitude and desire to make a purchase.

The study results derived from Indonesia provide a fascinating insight into the interplay between ESG factors, CSR, and CB. The research findings indicate that the incorporation of ESG elements in a company's operations has a favourable effect on its CSR efforts. However, it is observed that the influence of social and governance practices on CB is somewhat constrained. This suggests that Indonesian customers do not develop a distinct brand attitude or desire to make a purchase when firms communicate or promote their social and governance initiatives. Nevertheless, the research reveals a noteworthy revelation: the adoption of environmental practices has a substantial and favourable influence on the brand perception and purchase intention of Indonesian customers [45]. This finding is consistent with research published by Mandiri Bank, which emphasises that a significant proportion of individuals in Indonesia continue to possess an incomplete comprehension of the ESG framework. Another study also found that the demand for affordable products is prioritised over concerns about the environmental practices of corporations in developing countries such as Indonesia [46]. This helps explain their reluctance or indifference towards companies that adopt ESG and CSR strategies. The findings underscored the significance of environmental activities in influencing consumer behaviour, since it is the only dimension that Indonesian consumers are acquainted with and thus see as the most pivotal aspect [47].

### 5.2. Theoretical Implication

The study's theoretical implications contribute to the current body of knowledge on ESG and CSR policies. This study offers valuable insights that may be used to enhance corporate decision-making and policy development within this domain. This research aims to establish a connection between ESG factors and CSR, thus contributing to a deeper comprehension of their influence on customer behaviour. This theoretical contribution emphasizes the significance of integrating ESG practices and CSR into company strategy in order to address the changing expectations and needs of customers and improve marketing performance as a whole.

### 5.3. Managerial Implication

The study suggests a significant management conclusion, namely that organizations should prioritize good communication of their environmental policies in order to cultivate trust and bolster their reputation to the public. While listed companies are already required to comply with ESG requirements and reports, it would be beneficial to encourage non-listed companies to also recognise the importance of ESG and educate the public on how to interpret it. Leaving the responsibilities of ESG to only the listed companies

creates boundaries that overtime can belittle the importance to achieve sustainable goals. Policymakers can also support this by engaging in sufficient dialogue and taking initiatives to spread these concepts, ensuring that the public becomes increasingly familiar with them. They should also consider expanding the list of mandatory companies required to adopt ESG requirements. Policymakers should also implement policies to ensure that corporations that adopts ESG and CSR strategies does not exploit opportunities to greenwash.

Additionally, placing emphasis on and effectively communicating social and governance practices may be in line with customer expectations and have a favourable impact on consumer behaviour. By implementing a holistic approach to ESG and CSR policies, organizations have the potential to enhance their business performance, attain a competitive edge, and cultivate enduring sustainability and profitability. The study suggests a significant management conclusion, namely that organizations should prioritize good communication of their environmental policies in order to cultivate trust and bolster their reputation. Additionally, placing emphasis on and effectively communicating social and governance practices may be in line with customer expectations and have a favourable impact on consumer behaviour. By implementing a holistic approach to ESG and CSR policies, organizations have the potential to enhance their business performance, attain a competitive edge, and cultivate enduring sustainability and profitability.

The importance of this discovery is in its ramifications for the broader framework of CSR and environmental, social, and governance measures in Taiwan. The implication arises that the conventional governance structures and practices seen in family-owned enterprises can impede the efficient execution and assimilation of CSR endeavours. There exists a need to undergo a paradigm change in both attitude and strategy concerning corporate governance, particularly in the context of ESG-CSR initiatives. It is essential for Taiwanese firms to acknowledge the significance of implementing strong governance practices that foster transparency, accountability, and stakeholder engagement. By doing so, they can effectively increase their ESG initiatives and CSR endeavours, while also aligning with the expectations of diverse stakeholders and the broader community.

The results of this study have important implications for businesses operating in Indonesia, as they indicate that prioritizing environmental policies may have a major impact on customer attitudes towards brands and their desire to make purchases. Companies may boost their image and appeal to Indonesian customers by prioritizing and successfully promoting their environmental activities. Furthermore, the aforementioned results underscore the need for enhanced education and knowledge about the whole Environmental, Social, and Governance framework among consumers in Indonesia. This is crucial as it has the capability to influence their long-term attitudes and preferences.

*5.4. Limitations and Future Research Suggestion*

One potential constraint of this study is the dependence on self-reported data provided by the participants. Although self-reporting might provide helpful insights, it is crucial to recognize the possible presence of response bias. This implies that individuals may inadvertently provide erroneous or socially desirable replies, thus compromising the validity of the results. Moreover, it is noteworthy to mention that the research primarily concentrated on two nations, namely Taiwan and Indonesia. Although these nations provide useful insights, it is essential to acknowledge the possible constraints in extrapolating the results to other cultural or geographical settings. Given the limited availability of studies that explicitly differentiate between ESG and CSR, the authors have taken a meticulous approach in selecting research to safeguard the integrity of their study. Consequently, it is plausible that certain aspects may have eluded the analysis. However, despite these challenges, the findings of this research remain significant as they contribute valuable insights to the existing body of knowledge in this field.

Throughout the course of this research, it has become evident that Indonesians exhibit a limited level of familiarity with the intricate concepts of social and governance within the ESG framework [47]. Consequently, it is important to acknowledge that the findings of

this study may be influenced by their lack of awareness and understanding of the broader ESG concept.

Additionally, it is worth noting that the practice of ESG reporting in Taiwan is relatively new, as it only became mandatory for companies to disclose ESG reports in 2020. This means that there is less than 5 years of data available to study the ESG patterns in Taiwan. Although ESG is not a new concept to Indonesia, this research is subject to certain limitations due to the scarcity of relevant studies conducted in Indonesia, particularly surrounding on the variables of this research (ESG, CSR, and consumer behaviour). However, despite these limitations, the findings of this study provide valuable insights into the topic and contribute to the existing body of knowledge.

In order to overcome these constraints and bolster the reliability of the investigation, next research endeavours may consider investigating other methodologies, such as using objective metrics for data collection. This approach would provide a more impartial and dependable evaluation of the factors being examined. Overtime, researchers can gain deeper insights as they have access to a larger pool of materials to analyse. In addition, broadening the study's scope to include a wider array of nations will enhance the external validity of the results, facilitating a more thorough comprehension of the subject matter.

**Author Contributions:** Conceptualization, Y.H. and D.P.D.N.; methodology, Y.H.; validation, Y.H., C.H. and A.H.; formal analysis, Y.H.; investigation, D.P.D.N.; data curation, Y.H.; writing—original draft preparation, D.P.D.N.; writing—review and editing, Y.H., D.P.D.N., C.H. and A.H.; supervision, Y.H.; project administration, Y.H., C.H. and A.H.; funding acquisition, Y.H. All authors have read and agreed to the published version of the manuscript.

**Funding:** This research received no external funding.

**Informed Consent Statement:** Informed consent was obtained from all subjects involved in the study.

**Data Availability Statement:** All data are in the paper.

**Acknowledgments:** The authors would like to thank Yi Hsu for her unconditional support and extensive involvement in making this study happen. The authors would also like to thank their personal family members for the motivation. Lastly, the authors would like to express gratitude to the lab members for providing support in this study.

**Conflicts of Interest:** The authors declare no conflict of interest.

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
