# Peer review of "Investigating the Interconnection between Environmental, Social, and Governance (ESG), and Corporate Social Responsibility (CSR) Strategies: An Examination of the Influence on Consumer Behavior"

_sustainability, doi:10.3390/su16020614_

Round 1
Reviewer 1 Report
Comments and Suggestions for Authors
It is not clear how the authors measure individual components, e.g. CSR or S, E, G based on respondents' answers. Can this part of the methodology be supplemented?
Please also explain how the conclusion from the respondents' opinions was methodologically obtained: The study results indicate that the influence of ESG initiatives 548 on a company's CSR is not uniform across all initiatives. line 548-549
Author Response
We appreciate your comments and the responses as attached file.

Reviewer 2 Report
Comments and Suggestions for Authors
Main concepts. You will give more value to the paper if you provide clear conceptual definitions of ESG and CSR. The concepts you use interchangeably or imply a significant overlap.
At first, you used the abbreviation CB but the explanation is just in 12 page.
Methodology. Please provide a more detailed explanation of the research methodology, data collection, and analysis techniques. The use of broad survey methods and the reliance on self-reported data from a limited geographical region (Taiwan and Indonesia) can introduce biases and limit the generalizability of the findings.
References. The article should be better referenced, especially when the analysis of ESG, CSR, and consumer behavior is conducted.
The hypothesis should be better justified.
Hypothesis 1 (H1). Environment practices have a positive impact on company’s CSR. According title it should be strategies or practices?
Hypothesis 3 (H3). Social has a positive impact on a company’s CSR. Social what? Strategies or practices?
Do you analyse Environment or Environmental...????
Descriptive Statistics: The article extensively utilizes descriptive statistics without robust analytical or inferential statistical methods. Please think, about how to present the data in a more constructive way.
The limitations of the study are not adequately acknowledged or explored.
Generalizability and Contextual Relevance: The study's focus on specific geographic regions raises questions about its applicability to different cultural and economic contexts. The generalizability of the findings to broader global settings is thus questionable.
Practical Implications: While the study purports to offer practical implications for corporations, it fails to provide concrete, actionable recommendations. Authors should think about how to present this part better and emphasize practical relevance for business practitioners and policymakers.
To which part of the population do you draw the findings?
Comments on the Quality of English LanguageModerate editing of the English language required
Author Response
We appreciate your valuable comments and responses are as attached file.

Reviewer 3 Report
Comments and Suggestions for Authors
Dear Authors,
IThank you for your submission of the paper on the interconnection between ESG and CSR strategies. Your paper addresses a highly relevant and timely topic, aligning seamlessly with ongoing discussions about the implementation of sustainability-centered strategies in companies and the associated challenges.
One commendable aspect of your paper is its focused exploration of the subject matter, presenting a well-written review of the state of the art and incorporating key literature. The comprehensive coverage of current literature establishes a solid foundation for your research, providing readers with valuable context and insights.
The structure of the paper is well thought out, contributing to its overall readability. The decision to focus on Taiwan and Indonesia as case studies is well-documented, and your methodological approach appears suitable for the scope of the research. I found the presentation of results to be clear and thorough, and to the best of my knowledge, the applied methods appear to have been executed correctly.
The paper's strength lies in its detailed discussion and outlook, which enhances the overall quality of the research. The thoughtful analysis of the results, coupled with a forward-looking perspective, adds depth to the paper and provides readers with valuable implications for future considerations.
In summary, your submitted paper is not only interesting but also well-structured, focusing on an up-to-date and significant topic while effectively applying a case study approach. The combination of a thorough literature review, sound methodology, and insightful discussion makes your work a valuable contribution to the ongoing discourse on ESG and CSR strategies.
Author Response
We appreciate your comments and responses are as attached file.
